# Usefulness of the Modified Videofluoroscopic Dysphagia Scale in Determining the Allowance of Oral Feeding in Patients with Dysphagia Due to Deconditioning or Frailty

**DOI:** 10.3390/healthcare10040668

**Published:** 2022-04-01

**Authors:** Min Cheol Chang, Ho Yong Choi, Donghwi Park

**Affiliations:** 1Department of Rehabilitation Medicine, Yeungnam University Hospital, Daegu 42415, Korea; wheel633@ynu.ac.kr; 2Department of Neurosurgery, Kyung Hee University Hospital at Gangdong, Kyung Hee University College of Medicine, Seoul 05278, Korea; heoryong83@hanmail.net; 3Department of Physical Medicine and Rehabilitation, Ulsan University Hospital, University of Ulsan College of Medicine, Ulsan 44033, Korea

**Keywords:** dysphagia, aspiration pneumonia, penetration–aspiration scale, videofluoroscopic dysphagia scale

## Abstract

**Introduction**: In patients with dysphagia due to deconditioning or frailty, as with other disorders that cause swallowing disorders, the videofluoroscopic swallowing study (VFSS) is the gold standard for dysphagia evaluation. However, the interpretation of VFSS results is somewhat complicated and requires considerable experience in the field. Therefore, in this study we evaluated the usefulness of the modified videofluoroscopic dysphagia scale (mVDS) in determining whether to allow oral feeding in patients with dysphagia due to deconditioning or frailty. **Methods**: Data from the VFSS of 50 patients with dysphagia due to deconditioning or frailty were retrospectively collected. We evaluated the association between mVDS and the selected feeding method based on VFSS findings, and between mVDS and the presence of aspiration pneumonia. **Results**: Multivariate logistic analysis showed that the mVDS total score had a significant association with oral feeding methods based on VFSS findings in patients with dysphagia due to deconditioning or frailty (*p* < 0.05). In the receiver operating characteristic (ROC) curve analysis, the area under the ROC curve for the selected feeding method was 0.862 (95% confidence interval, 0.747–0.978; *p* < 0.0001). **Conclusions**: mVDS seems a valid scale for determining the allowance of oral feeding, and it can be a useful tool in the clinical setting and in studies that aim to interpret VFSS findings in patients with dysphagia due to deconditioning or frailty. However, studies involving a more general population of patients with dysphagia due to deconditioning or frailty are needed.

## 1. Introduction

Deconditioning is a complex process of physiological change following a period of inactive, sedentary or bed-rest lifestyle. It causes functional losses in such areas as mental status, and an inability to accomplish activities of daily living. It is frequently associated with hospitalization in the elderly [1]. The most predictable effects of deconditioning are seen in the musculoskeletal system, and include reduced muscle mass and strength. The decline in muscle mass and strength has been linked to falls, functional decline, increased immobility and frailty. This deconditioning also affects the swallowing muscles, which can lead to dysphagia.

Frailty is a complex syndrome that is characterized by the progressive decline in physical, mental, and social functions [2]. Frailty develops because of the age-related decline in several physiological systems [3]. It commonly occurs in older adults and deteriorates physiologic functions and increases the risk of poor health outcomes including disability, hospitalization, and mortality [3]. Frailty also leads to a decline in swallowing function [4].

Although the incidence of dysphagia in deconditioning or frailty has not been accurately reported, deconditioning and frailty have been reported in previous studies as among several causes of dysphagia [5,6,7,8]. When the degree of dysphagia is severe, oral feeding can result in aspiration pneumonia and mortality [9,10]. Therefore, in those with a high risk of aspiration pneumonia, oral feeding should be prohibited. However, because prohibiting oral feeding can lead to malnutrition, and a decline in physical function and general health, decisions regarding oral feeding should be carefully made after detailed ascertainment of its necessity [11].

The videofluoroscopic swallowing study (VFSS) is a standard tool for evaluating the severity of dysphagia, and it provides several details on swallowing function [12,13,14,15]. Based on VFSS results, clinicians can obtain information on dysphagia severity and develop a treatment plan [16]. However, the interpretation of VFSS results is somewhat complicated and requires experience in the field. Accordingly, objective criteria or cutoff values in diagnostic evaluation using quantitative analyses of VFSS should help clinicians decide on oral feeding more easily and precisely.

For the quantitative analysis of VFSS, the videofluoroscopic dysphagia scale (VDS) was developed, but it is limited because of the relatively low inter-rater reliability of some subcategories [17]. To compensate for these limitations, Chang et al. developed a modified version (mVDS) and demonstrated its high validation and inter-rater reliability [18]. mVDS includes nine parameters (lip closure, mastication, oral transit time, trigger pharyngeal swallow, epiglottis inversion, valleculae residue, pyriformis residue, pharyngeal wall coating, and aspiration) [18], and its clinical usefulness in evaluating swallowing function and selecting feeding methods was demonstrated in patients with amyotrophic lateral sclerosis and stroke, respectively [19,20].

Therefore, in the current study, we evaluated whether mVDS could be used to determine the possibility of oral feeding in patients with dysphagia due to deconditioning or frailty. Further, we investigated the mVDS cutoff value for permitting oral feeding.

## 2. Methods

### 2.1. Ethics Statements

This retrospective study was approved by the Institutional Review Board (IRB) of Ulsan University Hospital (2021-01-028). This study was performed according to the Declaration of Helsinki for human experiments. Informed consent was waived because of the retrospective nature of the study.

### 2.2. Study Design and Population

We retrospectively collected the data of patients with dysphagia due to deconditioning or frailty who had any symptoms of difficulty in swallowing and who underwent a VFSS at Ulsan University Hospital between April 2020 and January 2022. We obtained clinical data such as sex, age, presence of tracheal tube or Levin tube, and history of aspiration pneumonia.

The criteria for inclusion (patients with dysphagia due to deconditioning or frailty) were as follows: (1) age at VFSS > 50 years; (2) history of aspiration symptoms, such as coughing or choking; (3) presence of symptoms clinically indicative of dysphagia, such as reduced gag reflex or delayed swallowing reflex; (4) presence of medical records of hospitalization for medical reasons within the last year prior to VFSS (5) presence of chronic disease (hypertension, diabetes, or chronic kidney disease, etc.) (6) presence of oropharyngeal dysphagia due to deconditioning or frailty without a specific diagnosis that could cause dysphagia such as stroke, traumatic brain injury, or other laryngeal pathology [4].

The criteria for exclusion were as follows: (1) inability to sit, or difficulty maintaining consciousness; (2) dysphagia due to known neurologic conditions diagnosed by neurologists including stroke, traumatic brain injury, anoxic brain injury, brain tumor, motor neuron disease, Parkinson disease, or Alzheimer disease; and (3) dysphagia from laryngeal pathology, including laryngeal cancer, stenosis, paralysis, and postoperative head and neck surgery [4].

### 2.3. Aspiration Pneumonia

A retrospective review of medical records was performed to identify the development of aspiration pneumonia within one month before and after a VFSS in patients with dysphagia due to deconditioning or frailty. The following data were collected: symptoms, such as coughing during feeding; presence of sputum, dyspnea, or fever; chest radiography findings; blood laboratory findings (white blood cell [WBC] counts, C-reactive protein [CRP] level, and erythrocyte sedimentation rate); and use of antibiotics [21,22].

Although a definitive diagnosis of aspiration is difficult, and the diagnostic criteria for aspiration pneumonia are relatively different across studies, patients who met all of the following criteria were considered to have aspiration pneumonia in this study: (1) presence of both objective signs (coarse lung sounds, presence of lung infiltration on chest radiography, and systemic inflammation based on blood laboratory findings, such as increased CRP levels and WBC counts) and subjective symptoms (fever, cough, and increased purulent sputum), (2) clinical suspicion of aspiration (delayed swallowing or coughing during swallowing), and (3) no evidence of micro-organisms, such as *Legionella* or *Mycoplasma*, which are common pathogens in atypical pneumonia [21,22]. In addition, the clinical reports from the Department of Internal Medicine were used to diagnose aspiration pneumonia.

### 2.4. VFSS Protocol

VFSS was conducted with a fluoroscopic device and recorded as a video file. During the VFSS, patients consecutively swallowed the following materials, with a stepwise consistency: water, nectar (51–350 cP), rice porridge (351–1750 cP), and boiled rice (>1750 cP) [23]. The materials were mixed with liquid barium, and the patient swallowed them while in a relaxed sitting position. Dynamic fluoroscopic images were recorded in the anteroposterior and lateral views at 30 frames per second. The VFSS images were analyzed according to the Penetration–Aspiration Scale (PAS) and were considered positive for aspiration if the PAS score was >5 [24].

All studies were reviewed by two physiatrists who had at least eight years of experience in interpreting VFSS results. The conclusion (the allowance of oral feeding) was made by comprehensively considering the findings of VFSS, such as the oral transit time, the swallowing reflex, the VFSS results such as residues and aspiration, the amount of food swallowed, and the viscosity. Patient information, including age, sex, and underlying diseases, was withheld from the interpreters. The interpreters only observed the patients via the movie files on the laptop, described their findings, and chose a feeding method (non-oral feeding versus oral feeding) based on the VFSS results.

### 2.5. Modification of the VDS

As noted earlier, mVDS was developed based on a study regarding the inter-rater reliability of VDS. Among the VDS categories, the ones with a κ value < 0.2 (bolus formation, mastication, apraxia, tongue-in-palate contact, and pharyngeal transit time) were modified [25]. As mentioned by previous researchers, such categories had somewhat ambiguous guidelines, and three to four multiple selectable choices that led to low reliability [18,19,20]. Therefore, we modified the categories according to a binary scale or deleted the ambiguous categories. The mVDS was drafted as shown in Table 1. We also changed the category of laryngeal elevation to epiglottis inversion, which was reported to be an important factor in the swallowing process in a previous study, because laryngeal elevation and epiglottis inversion are a result of a combination of the contraction and relaxation of the suprahyoid and infrahyoid muscles [18,19,20]. To measure triggering pharyngeal swallow (swallowing reflex), the recorded fluoroscopic images in lateral view at 30 frames per seconds were analyzed by measuring the response time of pharyngeal swallowing reflexes. A study reported that, in the VFSS, the normal value of the response time of the swallowing reflex was 0.53 ± 0.64 s in elderly subjects (≥65 year old) [12]. If the response time of the swallowing reflex was longer than the normal value, we determined that the patient had a delayed swallowing reflex. Residues in the valleculae and pyriformis sinuses were divided into four groups: none, small (<10%), medium (10–50%), and large (>50%) [16].

Originally, to measure these VFSS findings as objective quantitative scores, VDS (with a sum of 100 points) was created based on the odds ratios of various prognostic factors. After modification of the VDS parameters, we rebalanced the score of each mVDS category, which yielded a sum of 100 points (Table 1) [18,19,20]. The inter-rater reliability (Cronbach α value) of the total score of the mVDS was 0.886, which was consistent with very good inter-rater reliability [20].

### 2.6. Statistical Evaluation

To find differences in parameters between the oral and non-oral feeding groups based on the VFSS findings, and between the aspiration and non-aspiration pneumonia groups, the Pearson chi-square test or Mann–Whitney U-test was used. Using only the parameters that were significant in the Mann–Whitney U-test, multivariate logistic regression analysis with the enter method was performed to identify parameters that had a significant association with the selected feeding method based on VFSS findings. To evaluate the accuracy of predictive factors for oral or non-oral feeding based on the VFSS findings, we performed receiver operating characteristic (ROC) analysis. Statistical analysis was performed using MedCalc (MedCalc Software, Ostend, Belgium) and SPSS version 22.0 (IBM Corp., Armonk, NY, USA).

## 3. Results

### 3.1. Patients’ Characteristics

Fifty patients with dysphagia due to deconditioning or frailty were enrolled in this study. Among them, 42 were male and 8 were female (mean age 71.26 ± 11.93 years). The patients’ demographic data are presented in Table 2.

### 3.2. Difference in Parameters between the Oral and Non-Oral Feeding Groups

In the Mann–Whitney U-test, age, PAS, and mVDS total score were significantly different between the oral and non-oral feeding groups (*p* < 0.05). In the multivariate logistic analysis, using these three parameters to identify any association with oral feeding methods based on VFSS findings, mVDS total score showed a significant association with oral feeding methods based on VFSS findings in patients with dysphagia due to deconditioning or frailty. (*p* < 0.05) (Table 3). In the ROC curve analysis, the area under the ROC curve for the selected feeding method was 0.862 (95% confidence interval [CI], 0.747–0.978; *p* < 0.0001). The optimal cutoff value for allowing oral feeding obtained from the maximal Youden index was ≤48 based on mVDS (sensitivity, 96.15%; specificity, 79.17%) (Figure 1).

### 3.3. Difference in Parameters between the Aspiration and Non-Aspiration Pneumonia Groups

In the Mann–Whitney U-test, age, PAS, and mVDS score were significantly different between the aspiration and non-aspiration pneumonia groups (*p* < 0.05). The multivariate logistic analysis using these three parameters to identify any association with the presence of aspiration pneumonia did not show any significant parameter.

## 4. Discussion

In this study, the mVDS score showed a statistically significant association with the allowance of oral feeding in patients with dysphagia due to deconditioning or frailty. To our knowledge, no previous study has investigated the usefulness of the mVDS score for evaluating dysphagia among patients with dysphagia due to deconditioning or frailty.

In mVDS, as in VDS, a higher score indicates a greater diet limitation and more severe dysphagia [18,19,20]. The results of this study suggest that an mVDS score of ≤48 is significantly associated with the allowance of oral feeding (sensitivity, 96.15%; specificity, 79.17%). Moreover, patients with an mVDS score of ≤12.5 showed 100% specificity for the allowance of oral feeding (sensitivity, 15.38%; specificity, 100.0%).

As mentioned earlier, the interpretation of VFSS results is somewhat complicated and requires considerable experience in the field, although VFSS is the gold standard for dysphagia evaluation because it is able to visualize the entire swallowing process, from the oral phase to the esophageal phase [12,13,14,15]. Therefore, numerous methods have been applied in interpreting VFSS results to quantify the severity of dysphagia and objectively determine the allowance of oral feeding based on the score. One of such methods is PAS, which is an eight-point scale used to describe the depth of food invasion and response to airway infiltration during VFSS [24]. Although it can notably define the extent of airway infiltration, the association between PAS score and feeding method based on VFSS results has been reportedly weak [20]. Another method to quantify dysphagia severity using VFSS video is VDS [25]. Previous studies have shown that VDS is significantly associated with penetration and/or aspiration occurring six months after the initial onset of dysphagia [25]. However, as previously mentioned, because of the ambiguity and complexity of some VDS subcategories, VDS has been reported to have low inter-rater reliability in such subcategories [25]. To improve these disadvantages, some VDS categories have been modified, and the application of these modifications in patients with stroke or other etiologies except dysphagia due to deconditioning or frailty was proven valid in previous studies [18,19,20]. In common with the results of these studies, our findings suggest that mVDS is a useful tool for quantifying the severity of dysphagia and determining the allowance of oral feeding in patients with dysphagia due to deconditioning or frailty.

## 5. Limitation

There are several limitations to our study. First, the total number of enrolled patients was relatively small. Therefore, generalizing our conclusions may be challenging. Studies with more patients are needed to achieve a more comprehensive conclusion. Second, this was a retrospective study; therefore, some data that might have added more value to the study were unavailable, such as frailty scores which can reflect the severity of disease at the time of VFSS. Lastly, this study was conducted in a single tertiary hospital, which might have also affected the generalizability of the study findings. Further large-scale prospective studies are necessary to achieve a comprehensive understanding of dysphagia due to deconditioning or frailty, excluding specific diseases that cause dysphagia.

## 6. Conclusions

mVDS seems a useful scale for determining the allowance of oral feeding and appears a relevant tool in the clinical setting, and in studies for interpreting VFSS findings in patients with dysphagia due to deconditioning or frailty. However, further studies involving a more general population of patients with dysphagia due to deconditioning or frailty are needed to elucidate a more accurate cutoff value for allowing oral feeding.

## Figures and Tables

**Figure 1 healthcare-10-00668-f001:**
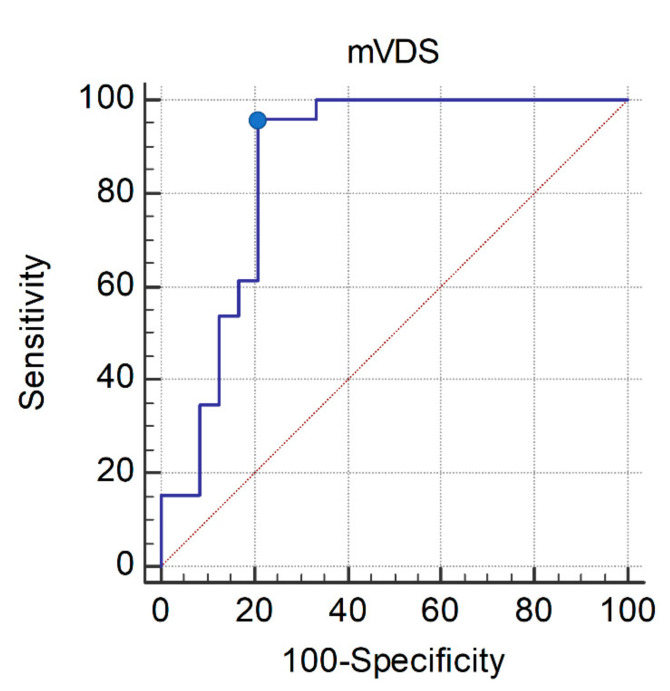
ROC curve of the mVDS score for the allowance of oral feeding in patients with dysphagia due to deconditioning or frailty. The optimal cutoff value (dots on the curves) of the mVDS score, which was obtained from the maximal Youden index, was ≤48 (AUC, 0.862; 95% CI, 0.747–0.978; *p* < 0.0001; sensitivity, 96.15%; specificity, 79.17%). ROC, receiver operating characteristic; AUC, area under the receiver operating characteristic curve; CI, confidence interval; mVDS, modified version of the videofluoroscopic dysphagia scale.

**Table 1 healthcare-10-00668-t001:** Modified version of the videofluoroscopic dysphagia scale.

Parameters		Score
lip closure	intact/not intact	0/6
massification	possible/not possible	0/11.5
oral transit time	≤1.5 s/>1.5 s	0/4
triggering pharyngeal swallow (swallowing reflex)	intact/delayed	0/7
epiglottis inversion	yes/no	0/13
valleculae residue	0%/<10%/≥10%, <50%/≥50%	0/3/6/9
pyriformis residue	0%/<10%/≥10%, <50%/≥50%	0/6.5/13/19.5
pharyngeal wall coating	no/yes	0/13
aspiration	intact/penetration/aspiration	0/8.5/17
total score		100

**Table 2 healthcare-10-00668-t002:** Characteristics of patients with dysphagia due to deconditioning or frailty in the present study.

Characteristics	Mean ± Standard Deviation (Minimum–Maximum)
age (year)	71.26 ± 11.93
sex (male:female)	42 (84.0%):8 (16.0%)
Tracheal tube (yes:no)	11 (22.0%):39 (78.0%)
Oral feeding: Levin-tube feeding	29 (58.0%):21 (42.0%)
History of aspiration pneumonia (yes:no)	25 (50.0%):25 (50.0%)
PAS grade	4.24 ± 3.24 (0–8)
mVDS scores	
lip closure	0.36 ± 1.44 (0–6)
mastification	0.92 ± 3.15 (0–11.5)
oral transit time	1.64 ± 2.04 (0–4)
triggering pharyngeal swallowing	3.87 ± 3.53 (0–7)
epiglottis inversion	2.38 ± 5.13 (0–13)
valleculae residue	5.76 ± 3.20 (0–9)
pyriformis residue	11.28 ± 7.08 (0–19.5)
pharyngeal wall coating	7.50 ± 6.73 (0–13)
aspiration	9.18 ± 7.45 (0–17)
total score	42.46 ± 21.70 (0–82.5)

PAS: penetration–aspiration scale, ALSFRS-R: revised amyotrophic lateral sclerosis functional rating scale, mVDS: modified videofluoroscopic dysphagia scale, MMSE: mini-mental status examination, MBI; modified Bathel Index.

**Table 3 healthcare-10-00668-t003:** Multivariate logistic regression analysis (with the enter method) of the association between the modified version of the videofluoroscopic dysphagia scale scores and the allowance of the oral feeding method.

Parameter	Beta Coefficient	Standard Error	OR (95% CI)	*p*-Value
Age	−0.088	0.049	0.916 (0.831–1.008)	0.073
PAS	−0.249	0.438	0.780 (0.331–1.839)	0.570
mVDS score	−0.071	0.034	1.081 (1.031–1.132)	0.020

PAS, penetration–aspiration scale; mVDS, modified version of the videofluoroscopic dysphagia scale; OR, odds ratio; CI, confidence interval.

## Data Availability

The data presented in this study are available on request from the corresponding author.

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
