# Peer review of "Usefulness of the Modified Videofluoroscopic Dysphagia Scale in Determining the Allowance of Oral Feeding in Patients with Dysphagia Due to Deconditioning or Frailty"

_healthcare, 2022, doi:10.3390/healthcare10040668_

Round 1
Reviewer 1 Report
Two reviewers in the last round of submission raised the major limitation that frailty was not defined, and therefore dysphagia associated with frailty specifically was not well established. The authors edits in this submission (highlighted in red on page 2) did NOT address this limitation. If you are investigating dysphagia in population X, how a participant belongs in population X is key to the methodology, and the authors have not explained how they did this. Inclusion criteria 4 is describing a process of elimination: you're ruling out disease causes of dysphagia, but not how it might be frailty related. Frailty has a clinically defined definition, and there are validated frailty scales in the literature that can help distinct individuals who are frail from those who are not. There is no evidence of this in this study, therefore there is no confidence that the participants in this study are actually frail - this is the major limitation. To address this, you can do one of two things:
1) if feasible, use information from the participant medical records to classify them as frail or not frail using a validated tool (e.g. the CSHA Clinical Frailty Scale)
OR
2) do not include frailty as a variable in the study; you can discuss the likelihood of frailty and its role in hospitalised patients in contributing to dysphagia, but it would not be a contributing factor in the cohort in this study. This option will also require a re-write of the introduction.
Additionally, the authors expressed the view that dysphagia due to frailty and dysphagia due to sarcopenia are similar, and while sarcopenia and frailty can overlap a great deal, sarcopenia dysphagia is specifically muscle related (which has been defined https://pubmed.ncbi.nlm.nih.gov/33620563/), while dysphagia due to frailty can have a wide array of causes because frailty is so multifactorial. It would be theoretically and clinically inaccurate to lump these two conditions together.
Other feedback:
Reference 6: is this the right reference?
I am recommending major revision as the authors seem to have misunderstood previous feedback and their author's response demonstrates a lack of understanding of the nature of the problem they claim to address. Actual edits required for the paper to be publishable does not have to be major if they understand the issues I have clarified in this review.
Author Response
Response to Reviewer`s Comments
Reviewer 1.
Two reviewers in the last round of submission raised the major limitation that frailty was not defined, and therefore dysphagia associated with frailty specifically was not well established. The authors edits in this submission (highlighted in red on page 2) did NOT address this limitation. If you are investigating dysphagia in population X, how a participant belongs in population X is key to the methodology, and the authors have not explained how they did this. Inclusion criteria 4 is describing a process of elimination: you're ruling out disease causes of dysphagia, but not how it might be frailty related. Frailty has a clinically defined definition, and there are validated frailty scales in the literature that can help distinct individuals who are frail from those who are not. There is no evidence of this in this study, therefore there is no confidence that the participants in this study are actually frail - this is the major limitation. To address this, you can do one of two things:
1) if feasible, use information from the participant medical records to classify them as frail or not frail using a validated tool (e.g. the CSHA Clinical Frailty Scale)
OR
2) do not include frailty as a variable in the study; you can discuss the likelihood of frailty and its role in hospitalised patients in contributing to dysphagia, but it would not be a contributing factor in the cohort in this study. This option will also require a re-write of the introduction.
Additionally, the authors expressed the view that dysphagia due to frailty and dysphagia due to sarcopenia are similar, and while sarcopenia and frailty can overlap a great deal, sarcopenia dysphagia is specifically muscle related (which has been defined https://pubmed.ncbi.nlm.nih.gov/33620563/), while dysphagia due to frailty can have a wide array of causes because frailty is so multifactorial. It would be theoretically and clinically inaccurate to lump these two conditions together.
Answer: We appreciate your valuable comment. We totally agree with your comment. It is also true that the definition of dysphagia due to frailty cannot define all patients enrolled in our study. Therefore, we have renewed the definition as “patients with dysphagia due to deconditioning or frailty” that can include all patients included in this study, and the introduction part has been significantly revised according to the reviewer's opinion. Indeed, deconditioning and frailty have been well known to cause dysphagia in several studies. The text has been revised to include this content.
Other feedback:
Reference 6: is this the right reference?
Answer: We appreciate your valuable comment. Reference 6 was deleted. Thank you.

Reviewer 2 Report
I understand that this paper is a study that researched the relationship between the VFSS results (whether patients were allowed to take orally) and the mVDS results by two evaluators. And you suggested that mVDS is effective. However, there are some points to clarify. Please refer to the following.
About the subject
Were these subjects hospitalized patients? Half of the subjects have a history of aspiration pneumonia, but what is the other half of the disease?
About Table 2
What did the subjects swallow to obtain these results?
VFSS results, such as oral transit time, swallow reflex, residue and aspiration, are affected by the combination of test food, swallowing volume, and swallowing posture, so mVDS and PAS results must specify these.
About Line 209-210 (Although it can...reportedly weak)
Please add references.
Author Response
Response to reviewer`s comment
Reviewer 2
I understand that this paper is a study that researched the relationship between the VFSS results
(whether patients were allowed to take orally) and the mVDS results by two evaluators. And you
suggested that mVDS is effective. However, there are some points to clarify. Please refer to the
following.
About the subject
Were these subjects hospitalized patients? Half of the subjects have a history of aspiration
pneumonia, but what is the other half of the disease?
Answer: We appreciate your valuable comment. In out study, we enrolled patients according to the inclusion & exclusion criteria previously described. We did not enroll patients based on specific VFSS findings. We do not include only hospitalized patients. All patients who underwent VFSS in our hospital were included, and among these patients, patients were included according to inclusion & exclusion criteria. In this study, we included patients with dysphagia caused by deconditioning or frailty, excluding various diseases that can cause swallowing disorders. Specific diseases were not considered for inclusion in this study.
About Table 2
What did the subjects swallow to obtain these results?
VFSS results, such as oral transit time, swallow reflex, residue and aspiration, are affected by the
combination of test food, swallowing volume, and swallowing posture, so mVDS and PAS results
must specify these.
Answer: We appreciate your valuable comment. mVDS is not determined by just one swallowing. As previously mentioned, during the VFSS, patients consecutively swallowed the following materials, with a step-wise consistency: water, nectar (51–350 cP), rice porridge (351–1,750 cP), and boiled rice (>1,750 cP). mVDS measures the worst score for each item among multiple swallowing processes in a patient and calculates the sum. As mentioned by two physiatrists who had at least 8 years of experience in interpreting VFSS results, the conclusion (the allowance of oral feeding) was made by comprehensively considering the oral transit time, the swallowing reflex, the VFSS results such as residues and aspiration, the amount of food swallowed, and the viscosity. In addition, mVDS and PAS are the tools used to measure the severity of dysphagia irrespective of these decisions. Following your comment, we have added it in manuscript as follows;
“All studies were reviewed by two physiatrists who had at least 8 years of experience in interpreting VFSS results. The conclusion (the allowance of oral feeding) was made by comprehensively considering the findings of VFSS, such as the oral transit time, the swal-lowing reflex, the VFSS results such as residues and aspiration, the amount of food swal-lowed, and the viscosity.”
About Line 209-210 (Although it can...reportedly weak)
Please add references.
Answer: We appreciate your valuable comment. Following your comment, we have added a reference in the manuscript.

Round 2
Reviewer 1 Report
It appears the authors have continued to misunderstand previous feedback about the issue of frailty in the studied cohort. The concern is that there is NO PROOF that the cohort is clinically frail; and with the new addition of deconditioning, there is also no proof that the cohort experienced deconditioning. Proving that this cohort was frail or was deconditioned is possible because both of these syndromes have clinical definitions and measures - but the authors have not provided any of this. It is possible that this cohort experienced frailty and deconditioning, but there is no proof in the data, therefore I can not accept that the data is about frail/deconditioned patients, and by extension the conclusions drawn can not be accepted for what the authors claim to be frail/deconditioned patients.
To be very clear, the motivation to validate the mVDS is fine, every other aspect of the study design is fine, the only issue is that the authors claim that the design targets frail/deconditioned participants but there is no definitive proof that they are frail/deconditioned. Either provide proof, or do not use these terms to classify the participants. I can not recommend acceptance of the manuscript in its current state.
Author Response
Reviewer 1
It appears the authors have continued to misunderstand previous feedback about the issue of frailty in the studied cohort. The concern is that there is NO PROOF that the cohort is clinically frail; and with the new addition of deconditioning, there is also no proof that the cohort experienced deconditioning. Proving that this cohort was frail or was deconditioned is possible because both of these syndromes have clinical definitions and measures - but the authors have not provided any of this. It is possible that this cohort experienced frailty and deconditioning, but there is no proof in the data, therefore I can not accept that the data is about frail/deconditioned patients, and by extension the conclusions drawn can not be accepted for what the authors claim to be frail/deconditioned patients.
To be very clear, the motivation to validate the mVDS is fine, every other aspect of the study design is fine, the only issue is that the authors claim that the design targets frail/deconditioned participants but there is no definitive proof that they are frail/deconditioned. Either provide proof, or do not use these terms to classify the participants. I can not recommend acceptance of the manuscript in its current state.
Answer: We appreciate your valuable comment. We totally agree with your opinion.
Basically, not all the patients in our study were conducted on inpatients, but all the patients included in this study had a history of hospitalization for medical diseases within one year prior to VFSS, and all of them had medical chronic diseases. To further clarify the inclusion criteria for patients with dysphagia due to deconditioning or frailty, we have added these to the text. Perhaps the addition of these inclusion criteria could better explain the inclusion target of this study as patients with deconditioning or frailty.
When we see patients with dysphagia in our dysphagia clinic, as in this study, we are faced with many patients with dysphagia along with the worsening of medical diseases without any special neurological disease such as stroke that can easily cause dysphagia. Although several previous studies have tried to define these patients as sarcopenic dysphagia, dysphagia due to deconditioning, or dysphagia due to frailty, it is true that there is no definitive diagnosis yet. However, due to the increase in life expectancy, the prevalence of these patients continues to increase, and it is now important to make definitions little by little through studies on these patients. We think that the definition of these patients should be clearly defined through further research or discussion at the dysphagia conference in the future. In addition, it is thought that various evaluation tools that can explain the severity and prognosis of dysphagia to these patients should be established through many studies or discussions at dysphagia conferences in the future.
Perhaps this study may be lacking as much as the reviewer's opinion, but we think it will be meaningful in showing the applicability of the evaluation tool called mVDS to these patients who can only be explained by frailty or deconditioning without such special neurological disease. We would like to thank you once again for your very detailed review.

Reviewer 2 Report
I appreciate your corrections.
The unclear points of the submitted paper were clarified by the corrections of the authors. I think this treatise deserves to be published in the journal.
Author Response
Reviewer 2
I appreciate your corrections.
The unclear points of the submitted paper were clarified by the corrections of the authors. I think this treatise deserves to be published in the journal.
Answer: We appreciate your valuable comment. Thank you.

This manuscript is a resubmission of an earlier submission. The following is a list of the peer review reports and author responses from that submission.
Round 1
Reviewer 1 Report
The authors described a new method to detect dysphagia in patients with frailty. The new score (mVDS) the authors invented was a relatively complicated one, since it needs volumes of residues and OTT measurement, as described below. So I think that a simplified one may be useful for clinical setting. There are also several points to be modified:
- This inclusion criteria enrolled relatively young adults (age > 20y). Frailty occurs frequently in individuals > 50 years. So exclusion of young adult may be desirable.
- The definition of frailty is not described.
- How they excluded Parkinson disease or Alzheimer disease was not clear. Are the authors board-certified neurologists?
- Was MMSE measured? If not, how is Alzheimer disease excluded?
- How was the triggering pharyngeal swallow defined in the method of VFSS? Please describe it in detail.
- How did they measure the volume of the valleculae or piriform residue in the method of VFSS? It may be highly subjective.
- What was the background disease or condition of patients with frailty? For example, polypharmacy or life-style related diseases.
- Inter-rater variation should be tested.
Reviewer 2 Report
This was a well written paper with clear aims and design. One major limitation with the manuscript is the focus on frailty without the support of diagnostic tools, please see my feedback below for details.
Section 2.2
Inclusion criteria: participants had to meet all inclusion criteria to be include din the study? E.g. participant who met all criteria but did not have a history of tube feeding would not be included? Table 2 indicates this is not the case, so you may need to revise how you've worded he inclusion criteria to be more clear.
Frailty: limitations section suggests no official frailty scales were used to make this diagnosis for participants. This is a major limitation in the methodology, as you are relying on a process of elimination (i.e. no other documented obvious causes) to determine frailty rather than actual diagnostic criteria. I would recommend that the authors edit the introduction where much focus has been placed on frail adults, and direct the focus to hospital inpatients as the studied cohort.
Section 2.5
line 124: epiglottis inversion instead of laryngeal inversion? The term used in the final mVDS uses "epiglottis inversion".
Table 2 title mislabelled as "ALS patients"? Legends also need some editing to match the reported variables.
Table 2 oral feeding: which one is yes which one is no? Please reorganise the table to improve labelling clarity, especially with the difference between categorical and continuous variables.
Section 3.2 Can you clarify whether the feeding method used as the outcome variable is based on the documented feeding method of the participants or as judged by the physiatrists? The implications are slightly different between these two variables, as one was the actual intake method of the participant, and the other one is the recommended intake method as perceived by the physiatrists. The two may overlap greatly and you may derive similar results in statistical analysis, but it's important to make the distinction.
Additionally, please report the inter-rater reliability of the mVDS between the two physiatrists who rated the VFSS, especially as this was an issue with the VDS.
Reviewer 3 Report
This is a retrospective review of patients with frailty and the presence or absence of aspiration pneumonia. A modified Videofluoroscopic Dysphagia Scale was employed to predict those patients who could be cleared for oral diet. A score of less than 48 was sensitive for 96% of those who were able to have oral feeding. A score of 15 or less had 100% specificity.
The methodology seems sound and the predictive value of PAS in univariate analysis seems to confirm that it is a reasonable patient selection.
I would modify the appearance of table 1 and 2, to allow to read across the columns more easily, adding lines perhaps.
Their statistical analysis seems valid, though increasing patient population size was one of the limitations they mentioned in the paper.
One of the strengths of the paper is that they are trying to focus on patients who have isolated frailty, though that may be difficult to parse out for the clinician who is seeing a patient with multiple comorbidities.